# Genomic epidemiology of *Vibrio cholerae* reveals the regional and global spread of two epidemic non-toxigenic lineages

Haoqiu Wang[1], Chao Yang[2]*, Zhou Sun[3], Wei Zheng[1], Wei Zhang[1], Hua Yu[1], Yarong Wu[2], Xavier Didelot[4,5], Ruifu Yang[2], Jingcao Pan[1]*, Yujun Cui[2]*

**1** Microbiology Laboratory, Hangzhou Center for Disease Control and Prevention, Hangzhou, Zhejiang Province, China, **2** State Key Laboratory of Pathogen and Biosecurity, Beijing Institute of Microbiology and Epidemiology, Beijing, China, **3** Institution of Infectious Disease Control, Hangzhou Center for Disease Control and Prevention, Hangzhou, Zhejiang Province, China, **4** School of Life Sciences, University of Warwick, Gibbet Hill Campus, Coventry, United Kingdom, **5** Department of Statistics, University of Warwick, Coventry, United Kingdom

☯ These authors contributed equally to this work.
* yangchao2835@gmail.com (CY); jingcaopan@sina.com (JP); cuiyujun.new@gmail.com (YC)

## Abstract

Non-toxigenic *Vibrio cholerae* isolates have been found associated with diarrheal disease globally, however, the global picture of non-toxigenic infections is largely unknown. Among non-toxigenic *V. cholerae*, *ctxAB* negative, *tcpA* positive (CNTP) isolates have the highest risk of disease. From 2001 to 2012, 71 infectious diarrhea cases were reported in Hangzhou, China, caused by CNTP serogroup O1 isolates. We sequenced 119 *V. cholerae* genomes isolated from patients, carriers and the environment in Hangzhou between 2001 and 2012, and compared them with 850 publicly available global isolates. We found that CNTP isolates from Hangzhou belonged to two distinctive lineages, named L3b and L9. Both lineages caused disease over a long time period with usually mild or moderate clinical symptoms. Within Hangzhou, the spread route of the L3b lineage was apparently from rural to urban areas, with aquatic food products being the most likely medium. Both lineages had been previously reported as causing local endemic disease in Latin America, but here we show that global spread of them has occurred, with the most likely origin of L3b lineage being in Central Asia. The L3b lineage has spread to China on at least three occasions. Other spread events, including from China to Thailand and to Latin America were also observed. We fill the missing links in the global spread of the two non-toxigenic serogroup O1 *V. cholerae* lineages that can cause human infection. The results are important for the design of future disease control strategies: surveillance of *V. cholerae* should not be limited to *ctxAB* positive strains.

## Author summary

Non-toxigenic *Vibrio cholerae* isolates are associated with diarrheal disease globally. Among them, *ctxAB* negative, *tcpA* positive (CNTP) isolates have the highest risk of

**Data Availability Statement:** The genomes of all the analyzed strains are available in GenBank under the accession numbers listed in S1 Table. Newly

sequenced genomes are available under the BioProject ID PRJNA492763.

**Funding:** This work is supported by the National Key Research & Development Program of China (No. 2018YFC1603902), National Key Program for Infectious Diseases of China (No. 2018ZX10101003 and 2018ZX10714-002), National Natural Science Foundation of China (No.81071395), State Key Development Program for Basic Research of China (No. 2015CB554202), and Hangzhou key medicine discipline fund for public health laboratory sponsored by the Hangzhou government (2016-2018). The funders had no role in study design, data collection, data analysis, interpretation, and writing of the report.

**Competing interests:** The authors declare that they have no competing interests.

disease because they may be able to colonize the human intestine. By population genomic analysis of 850 previously published genomes and 119 newly sequenced genomes in this study, we found that most of CNTP isolates can be attributed into two distinctive lineages, L3b and L9. Both lineages are not only circulating in local regions causing endemic disease, but have been spreading worldwide over the past 100 years. Central Asia is the most likely origin of the L3b lineage, and the strains spread to China on at least three occasions. The L3b strains were also exported from China to other parts of the world, and one of the migrations are related to the groups that circulated in Latin America. Within Hangzhou, a modernized city in China, aquatic food products are the most likely medium of CNTP strains, and a spread pattern from rural aquafarms to the urban areas was observed. This work filled the missing links in the global spread of two non-toxigenic serogroup O1 *V. cholerae* lineages that can cause human infection. The results suggest that in future disease control efforts, the sampling of *V. cholerae* should not be limited to the apparently toxic isolates, and that a more thorough and unbiased sampling framework is needed.

## Introduction

*Vibrio cholerae* is the causative agent of cholera, an acutely dehydrating diarrheal disease that can kill its victims within hours if left untreated [1]. It is estimated that each year there are 1.3 to 4 million cholera cases, resulting in 21,000 to 143,000 deaths worldwide [2]. *V. cholerae* has been classified based on the surface somatic O antigens and more than 200 serogroups are identified to date [3]. Cholera epidemics are caused by isolates of serogroups O1 and O139, with O1 being further differentiated into two biotypes, classical (Cla) and El Tor (ET) [4]. There have been seven historical cholera pandemics since 1817. The Cla biotype is believed to have caused the first six pandemics, whereas the ET biotype replaced Cla globally to cause the seventh cholera pandemic that has been ongoing since 1961 [5]. O139 isolates were first identified in India and Bangladesh in 1992, which were found to be derived from the ET biotype and have not spread beyond Asia [6].

Whole-genome sequencing provides the highest possible discrimination power between bacterial isolates, and has been put to great use in *V. cholerae* research, for example to track the source of the 2010 Haiti cholera outbreak [7] and to reconstruct the global transmission routes of the seventh pandemic [5,7–10]. Genome-wide single nucleotide polymorphism (SNP) analysis showed that *V. cholerae* can be divided into eight major lineages (L1-L8) [5]. Cla isolates belong to lineage L1, and ET isolates are assigned to five lineages, of which L2 is responsible for the ongoing seventh pandemic, while the others are geographically restricted to specific regions and can cause sporadic cholera cases (L3 in US Gulf Coast, L5 in Sulawesi, L6 in Saudi Arabia, and L8 in Australia). L4 and L7 isolates are distinct from the other lineages and are usually non-pathogenic and isolated from the environment [5].

Despite the high genetic diversity of *V. cholerae*, only lineages encoding the key virulence factors can cause cholera pandemics (e.g. L1 and L2). Cholera toxin (CT) and toxin co-regulated pilus (TCP) are well-known virulence factors. CT, the protein complex that causes the dehydration and watery diarrhea, is encoded by *ctxA* and *ctxB*, which are located in a temperate filamentous bacteriophage CTXφ [11]. When it is released and enters epithelial cells, it induces a rapid and massive loss of body fluids, which is the primary cholera symptom. The TCP gene cluster is part of a *Vibrio* pathogenicity island (VPI) called VPI-1 and responsible for attachment to the host intestinal epithelium, where it serves as the receptor of CTXφ [12].

The major subunit of TCP is encoded by *tcpA* [13], which is the most diverse gene [14] in the TCP gene cluster.

CT producing *V. cholerae* can cause worldwide cholera epidemics, in contrast, non-toxigenic (*ctxAB* negative) isolates tend to be associated with sporadic localized diarrheal disease [15,16]. For instance, non-toxigenic isolates have been found associated with sporadic human infection in Latin America (serogroup O1 isolates) [8,17,18], Thailand (O27) [19], Iraq (O53) [20], and Japan (O48) [21], and reported to cause small-scale local outbreaks (less than 30 cases with O1) in India [22], Uzbekistan [23], Russia [24,25], Fiji [26], and China [27,28] between 1965 and 2014. Multi-locus sequence typing data revealed that some non-toxigenic clonal complexes were geographically widespread [16], however, the global picture of non-toxigenic infections is largely unknown.

One concern of non-toxigenic isolates is that whether they can gain *ctxAB* genes and cause epidemic cholera. However, most of the non-toxigenic isolates do not carry the *tcpA* gene [16], indicating a small probability of colonizing the human intestine. In contrast, *ctxAB* negative, *tcpA* positive (CNTP) isolates have higher potential risk of disease outbreak among the non-toxigenic isolates. From 2001 to 2012, a total of 71 infectious diarrhea cases caused by CNTP serogroup O1 *V. cholerae* isolates were confirmed in Hangzhou, an East China city which is 50 kilometers from the Western Pacific. In order to characterize the CNTP isolates and the human infection they caused, we sequenced the whole genomes of 119 Hangzhou isolates, including 91 CNTP isolates (71 from patients, 13 from carriers and 7 from the environment) and 28 contemporaneous non-CNTP strains, and compared them with 850 publicly available *V. cholerae* genomes representing the global diversity of the species.

## Materials and methods

### *V. cholerae* isolates

According to the laws of the People's Republic of China on Prevention and Treatment of Infectious Diseases, cholera is an infectious disease under Class A, and all suspected cholera patients admitted to hospitals need to be tested for *V. cholerae*. The samples of suspected patients were sent to Hangzhou CDC for laboratory confirmation. Stool specimens of suspected patients were cultured in the selective No. 4 agar for *V. cholerae* isolation and identification. If the culture result was positive for *V. cholerae*, the patient was treated as a cholera case, and the corresponding epidemiological investigation was initiated to collect samples from people, food and water that had been exposed to the patient in the five days prior to disease onset. If the food sample was positive for *V. cholerae*, the source of the food was further traced. If the person who had been in contact with the patient was asymptomatic but the stool specimen was positive for *V. cholerae*, the person was defined as a carrier. Infectious diarrhea cases caused by CNTP isolates were treated in the same way as cholera cases, and the epidemiological investigation was consistent with that of cholera. In total 91 CNTP strains (all belonging to serogroup O1) were isolated during 2001–2012, representing 81% of the *V. cholerae* serogroup O1 strains (12% were CPTP and 7% were CNTN) isolated in Hangzhou during the same period. Among the CNTP strains, 71 were isolated from patients, 13 were isolated from carriers and seven were isolated from the environment (aquatic animal, water or tableware). 28 other representative isolates, including 13 CNTN (*ctxAB⁻*, *tcpA⁻*, serogroup O1) and 15 CPTP (*ctxAB⁺*, *tcpA⁺*, eight serogroup O1 and seven O139 strains) strains that were isolated during the same period, were used for comparison in this study (S1 Table). The criteria used for clinical symptom classification is listed in S2 Table.

To determine the potential gene loss during laboratory culture, the presence/absence of *ctxAB*, *zot*, *ace*, *cep* and *tcpA* were examined by PCR amplification within three days of

isolation (most were tested within a day). In a previous study, it has been reported that *ctxAB* genes of *V. cholerae* isolates can be lost in culture, however, the earliest loss took place four days after isolation and the loss was not only the *ctxAB* genes but also the other genes of the CTX prophage [29]. Thus, although the *ctxAB* genes are on a mobile element, the probability of loss during laboratory culturing should be small for our collection.

To exclude the possibility of mixed infection of *ctxAB* positive and negative strains, we selected three colonies for each sample to test the presence/absence of *ctxAB* by PCR amplification separately. No sample showed co-existence of both *ctxAB* positive and negative colonies in all 119 Hangzhou samples, which was consistent with a previous report of low within host diversity of *V. cholerae* (zero to three SNPs within each patient) [30]. Finally, the samples of *ctxAB* negative and *tcpA* positive were defined as CNTP strains.

## Antimicrobial susceptibility testing

The antimicrobial susceptibilities of 22–80 randomly selected Hangzhou CNTP isolates were tested using the disk diffusion method on Mueller-Hinton agar according to the guidelines of the Clinical and Laboratory Standards Institute (CLSI, M02-A11). The isolates were categorized as being resistant, intermediate and sensitive to each antimicrobial drug. The susceptibilities to drugs, including ampicillin, amikacin, streptomycin, gentamicin, tetracycline, chloramphenicol, trimethoprim-sulfamethoxazole (SXT) ciprofloxacin, and nalidixic acid, were interpreted (S3 Table) using the CLSI criteria (M100-S22) for the *Enterobacteriaceae*.

## Genome dataset

A total of 969 *V. cholerae* genomes were used in this research, including 119 newly sequenced genomes of Hangzhou isolates, and 850 publicly available genome sequences downloaded from NCBI (up to July 2018). The 850 public genomes were collected for various purposes and do not represent a well-defined epidemiological cohort. They were isolated from various sources during 1937–2018, and covered 50 countries on six of the seven continents (except Antarctica). The genomes of all the analyzed strains are available in GenBank under the accession numbers listed in S1 Table, newly sequenced genomes are available under the BioProject ID PRJNA492763.

## Genome sequencing and variant calling

The DNA of Hangzhou isolates was extracted using DNeasy Blood & Tissue kit (QIAGEN), and subjected to paired-end library (average insert size of 350 bp) preparation using the NEBNext Ultra DNA Library Prep Kit (NEB). Whole genome sequencing was performed using Illumina Hiseq 4000 platforms, the read length is 150 bp and 400 Mb clean data were generated for each strain on average.

We performed *de novo* assembly using SOAPdenovo v2.04 [31] as previously described [32]. The number of contigs and average size of assemblies were 163 (46–495, > 500 bp) and 4.0 Mb (3.9–4.3), with an average of 81-fold (54–96) depth for each *V. cholerae* genome. SNPs were identified as previously described [32,33]. Firstly, the assemblies were aligned against the reference genome using MUMmer v3.1 [34], to generate the whole genome alignment and identify SNPs in the core genome (regions present in all isolates). Secondly, the clean sequencing reads of Hangzhou isolates were mapped to the assemblies to evaluate the SNP accuracy using SOAPaligner [35], only high-quality SNPs (supported by >10 reads, quality value >30) were kept in further analysis. The repetitive regions of the reference genome were identified using TRF v4 [36] and BLASTN search against self. SNPs in repetitive regions were excluded and only bi-allelic SNPs were used in further analysis. N16961 (NC_002505.1, NC_002506.1)

was used as the reference sequence in the analysis of 969 genomes, a total of 562,172 SNPs (2.16 Mb core genome) were identified from the 969 isolates, which were used to construct the maximum likelihood tree using FastTree V2 [37]. Phylogenic analysis showed that the Hangzhou isolates belonged to two distinctive lineages, which were named L3b and L9 (Figs 1A and S1).

The core regions of different genome sets are variable and this affects SNP calling and phylogenetic reconstruction. To obtain a high phylogenetic resolution, we re-selected reference sequences for L3b and L9 lineages to improve the whole genome alignments. Strain MS6 (NZ_AP014524.1, NZ_AP014525.1) was used as the reference sequences for L3b isolates, 40,533 SNPs (3.52 Mb core genome) were identified for all the 119 L3b isolates, 27,006 SNPs (3.68 Mb core genome) were identified for the 81 Hangzhou L3b isolates. Strain 2012Env-9 (NZ_CP012997.1, NZ_CP012998.1) was used as the reference sequences for L9 isolates, 18,937 SNPs (3.76 Mb core genome) were identified for all the 20 L9 isolates, 19,874 SNPs (3.83 Mb core genome) were identified for the 11 Hangzhou L9 isolates.

## Phylogenetic analyses

We used ClonalFrameML [38] to infer the recombination events in L3b and L9 lineages as previously described [39]. The input maximum likelihood trees were constructed using FastTree V2 [37] based on the whole genome alignments, non-core sites were ignored during the calculation. ClonalFrameML analysis revealed that the recombination rates of L3b and L9 lineages were much higher than that of L2 pandemic lineage (S2 Fig). The ratio of effects of recombination and mutation (*r/m*) of L3b lineage was 36.1, with ratio of recombination to mutation rate (*R/theta*) of 0.52, mean DNA import length (*delta*) of 4,082 bp, and mean divergence of imported DNA (*nu*) of 0.017. The *r/m* of L9 lineage was 23.7, with *R/theta* = 0.67, *delta* = 2,716 bp and *nu* = 0.013. In contrast, the recombination rate of L2 pandemic lineage was much lower, with *r/m* previously reported to be 0.47 [10]. We also ran Gubbins [40] to identify the recombination regions in the L3b and L9 lineages. The recombination regions identified by the two methods were highly consistent, with almost all the recombination regions (1.868/1.872 Mb in L3b lineage, 1.065/1.070 Mb in L9 lineage) identified by ClonalFrameML being also found in Gubbins results, further supporting the inference of high recombination rates of these two lineages.

SNPs located in inferred recombined regions identified by ClonalFrameML were excluded prior to further phylogenetic analysis. After removing the recombinant SNPs, 535 SNPs (1.56 Mb non-recombined core genome) for L3b isolates, 662 SNPs (2.23 Mb non-recombined core genome) for Hangzhou L3b isolates, 509 SNPs (2.66 Mb non-recombined core genome) for L9 isolates and 448 SNPs (2.72 Mb non-recombined core genome) for Hangzhou L9 isolates were used to construct the maximum likelihood trees using RAxML v8 [41] under the GTRGAMMA model with 100 bootstraps (Figs 1B, 1C, 2B and 2C). All the phylogenetic trees were visualized using ggtree v1.16 [42].

## Presence/absence of virulence factors

We examined the sequences of 17 regions that were previously described as associated with pandemic disease [7,43], including five major pathogenicity islands (CTXφ, VPI-1, VPI-2, VSP-1 and VSP-2), 10 genomic islands (GI-1~10), the SXT region and the superintegron region (S3 Fig). The sequences of N16961 were used as the reference sequences except for SXT, which is absent from this genome. The SXT region sequence of MJ-1236 (NC_012668.1, NC_012667.1) was used as the SXT reference sequence. The sequences of these genome elements were split into segments of 500 bp, and BLASTN was used to examine the presence/

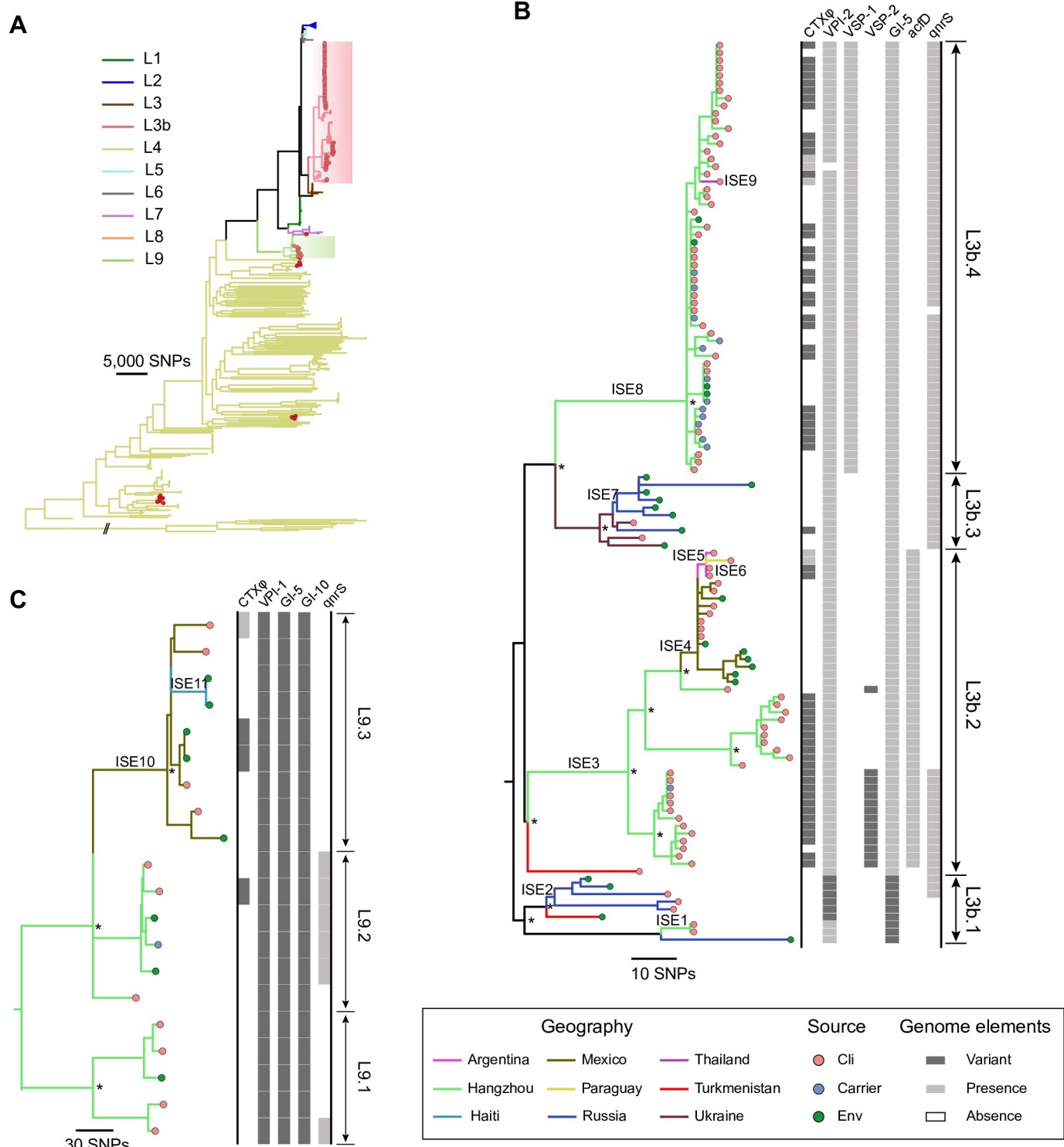

**Fig 1. Phylogeny of global 969 *V. cholerae* isolates (A), L3b (B) and L9 (C) isolates.** (A) Maximum likelihood tree of global 969 *V. cholerae* isolates. Branch colors indicate the lineages, and red circles in the tree tips indicate Hangzhou isolates. L2 lineage isolates were collapsed for visualization. (B, C) Maximum likelihood trees of L3b (B) and L9 (C) lineages based on non-recombined SNPs. L3b.1 and L9.1 isolates were used to root the trees based on Figs 1A and S1. Branch colors indicate the geographical locations, and circles in the tree tips indicate the sources, red for clinical (Cli) isolates, blue for carrier and green for environmental (Env) isolates. Presence/absence of genomic elements are shown on the right, white for absence, grey for presence of El Tor elements and dark grey for presence of non-El Tor variants. The variant of CTXφ indicates the pre-CTXφ that lacking *ctxAB* genes. A hundred iterations of bootstrap were performed in (B, C) and asterisks indicate that bootstrap values of nodes are greater than 80.

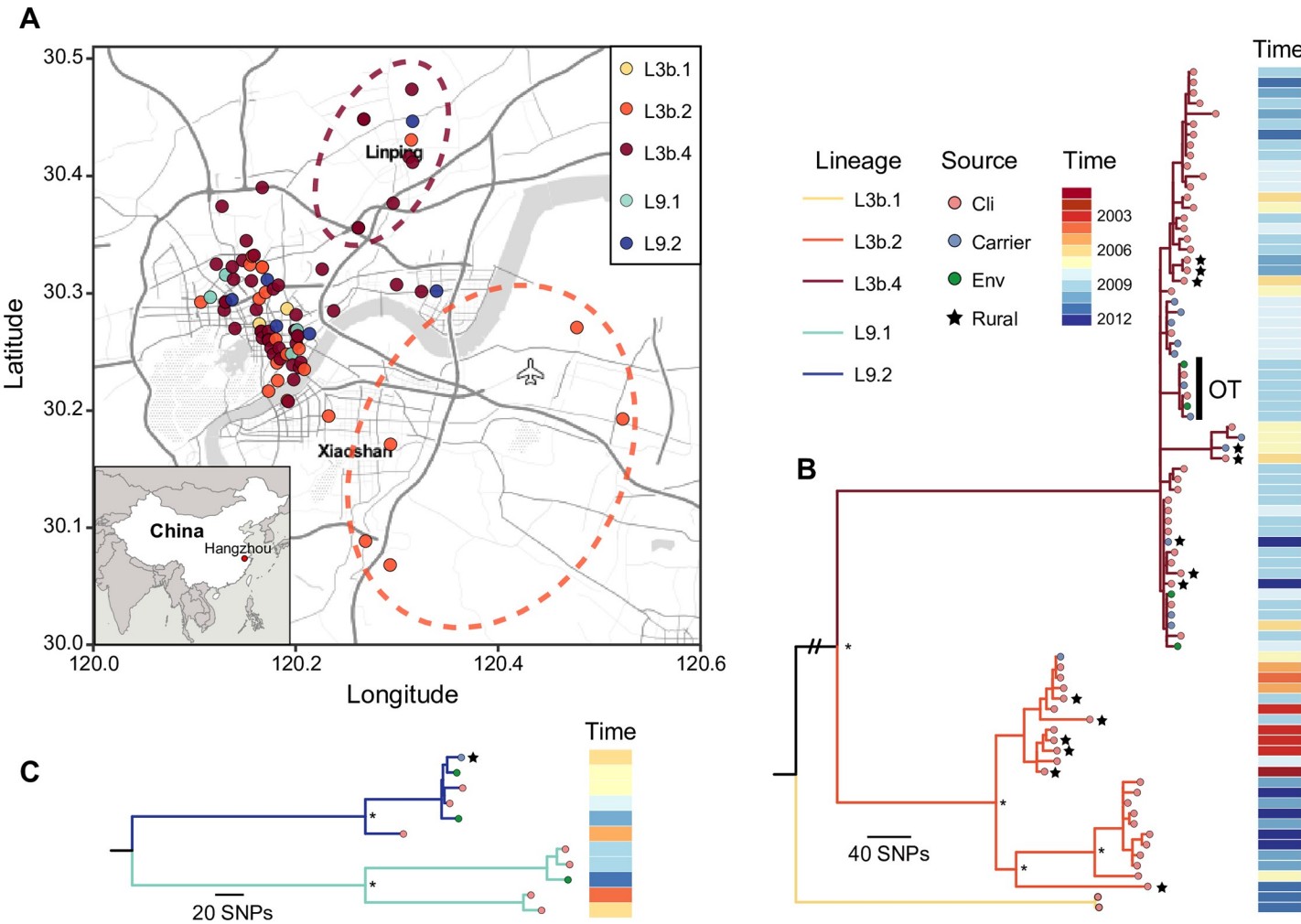

**Fig 2. Local spread of L3b and L9 lineages in Hangzhou.** (A) Geographical distribution of Hangzhou L3b and L9 lineage isolates. Circle colors indicate sub-lineages. Dotted ellipses indicate possible rural sources of L3b.2 and L3b.4, the points within the dotted ellipses indicate rural isolates, which were represented by stars in panel B and C. The map was created using ggmap [58] based on the public geographical data downloaded from OpenStreetMap. (B, C) Maximum likelihood trees of Hangzhou L3b (B) and L9 (C) isolates. L3b.1 and L9.1 isolates were used to root the trees and are the same as in Fig 1B and 1C. Branch colors indicate sub-lineages, and circles in the tree tips indicate sources. Stars indicate rural samples. Isolates of a confirmed outbreak event is marked by a vertical line and labeled "OT" (outbreak). Isolation times are shown on the right. Asterisks indicate that bootstrap values of nodes are greater than 80 (100 bootstraps). Double slash indicates artificially shortened branch.

absence of each region in each genome. We also searched against VFDB (virulence factor database) [44] and ARDB (antibiotic resistance genes database) [45] to identity possible virulence and antibiotic genes. A gene/segment was considered present if the overall hit coverage and identity were at least 70%.

## Hierarchical clustering analysis

We performed clustering analysis for all the 969 isolates, L3b and L9 lineage isolates using the hierarchical Bayesian Analysis of Population Structure (hierBAPS) [46]. Firstly, hierBAPS was run on the 562,172 SNPs of 969 isolates with the maximum number of populations ($K$) of 100, 200 and 300. 969 isolates were clustered into five ($K$ = 100 and 200) or nine ($K$ = 300) lineages (S1 Fig). Lineage L2, L3, L3b, L5, L6 and L8 were clustered into hierBAPS cluster C1, lineage L1, L7 and L9 were clustered into hierBAPS cluster C2, and lineage L4 were split into three or

seven hierBAPS clusters. HierBAPS clusters generally contain more isolates than that of L1-L9, and no CNTP isolates specific clusters were identified. To focus on the CNTP isolates and keep the unity to previous study [5], we used the designation of lineage L1-L9 in this study.

HierBAPS was then run on the non-recombined SNPs of L3b and L9 lineages, with the maximum *K* of 5, 10 and 15. L3b lineage isolates were split into four or six hierBAPS clusters, and L9 lineage isolates were split into 3 clusters which are named L9.1-L9.3 sub-lineages (S4 Fig). For L3b lineage, hierBAPS cluster C2 (*K* = 5) were split into three clusters (C5-C7) when *K* was set to 10 and 15, however, in the phylogenetic tree, it was clear that isolates of C5-C7 have a common ancestor, which were supported by a long common branch with a high bootstrap value (100). Combining the hierBAPS clustering and phylogenic tree, we designated L3b lineage into four sub-lineages, L3b.1-L3b.4. Additionally, one hierBAPS cluster C1 isolate M229, clustered together with L3.2 (C2 or C5-7) isolates in the phylogenetic tree, which were supported by a high bootstrap value (100), we assigned it into L3.2 sub-lineage.

## Molecular clock analysis

We investigated the temporal signal of L3b and L9 lineages using TempEst v1.5 [47], by calculating the linear regression between root-to-tip distance and isolation date. Three date-randomization tests were performed using TipDatingBeast [48], and we found a significant temporal signal in the L3b lineage (S5 Fig). The 118 L3b isolates with known isolation dates were further used in BEAST v1.10 [49] to date the common ancestors. 535 non-recombinant SNPs were included in this analysis, using the GTR + Γ substitution model, isolation years for tip date calibration and constant sites correction. We tested eight combinations of molecular clock and tree models to identify the best fit, three independent chains for each combination were run to ensure convergence. Both path sampling and stepping-stone sampling indicate the best fit was an uncorrelated relaxed clock with a Bayesian skyline coalescent tree (S4 Table), which were used as the priors of further analysis. We ran three independent Markov chain Monte Carlo chains over 100 million steps, sampling every 2,000 steps. Tracer v1.7 [50] was used to read the BEAST outputs and calculate the effective sample size (ESS) values, and the ESS of parameters of all three runs were greater than 200. The tree estimates of three runs were combined using LogCombiner v1.10 and were used to generate the maximum clade credibility trees (S6 Fig) using TreeAnnotator v1.10, with the first 10% of states removed as burn-in. The BEAST estimated molecular clock of L3b lineage strains was $3.73 \times 10^{-7}$ substitutions per site per year, with 95% confidence interval (CI) $2.78–4.72 \times 10^{-7}$, which is lower than the previously estimated mutation rate of the seventh pandemic group: $6.1 \times 10^{-7}$ (95% CI: $5.7–6.4 \times 10^{-7}$) substitutions per site per year [9].

## Ethics statement

Ethical approval was not required because this research was done as part of routine infectious disease surveillance and management.

## Results

### *V. cholerae* lineages L3b and L9 caused epidemics of diarrhea in Hangzhou

We constructed a maximum likelihood (ML) tree of 969 isolates based on genome-wide SNPs, including 119 Hangzhou isolates and 850 global isolates (Figs 1A and S1, S1 Table). The isolates from Hangzhou showed a remarkably high diversity, which can be attributed to five lineages. Most Hangzhou CNTP isolates (80/91) belonged to an undefined lineage that was closely related to US Gulf Coast L3 isolates [5,51], and was named L3b in this study. The remaining 11

CNTP isolates were attributed to another lineage we named L9, which was phylogenetically intermediate between lineages L4 and L1. Two Hangzhou CPTP isolates also belonged to lineage L3b and the remaining 26 isolates were assigned to pandemic lineages L2 (n = 12, all CPTP), L4 (n = 13, all CNTN) and L7 (n = 1, CPTP). In addition, 24 Latin American isolates were assigned to L3b and L9 lineages, which were previously considered as local isolates and named as MX-2 and ELA-3 lineage, respectively [8].

To describe the phylogenetic relationships with greater resolution, we re-examined the core genomes of L3b and L9 isolates, removed the recombined regions and constructed ML trees separately based on new SNP sets (Fig 1B and 1C). The ML trees further divided L3b into four sub-lineages, L3b.1~4, and divided L9 into three sub-lineages, L9.1~3. We found that Hangzhou L3b isolates were concentrated on L3b.2 (24/82) and L3b.4 (56/82), except for two closely related isolates that belonged to L3b.1, while L3b.1 and L3b.3 included isolates from Turkmenistan, Russia, and Ukraine. Hangzhou L9 isolates were concentrated on L9.1 (5/11) and L9.2 (6/11), whereas the L9.3 sub-lineage was composed of Mexico and Haiti isolates.

## Clinical characters and epidemiological patterns of L3b and L9 isolates

The clinical symptoms caused by L3b and L9 isolates were mostly mild or moderate (66/67, S7A Fig). Patients usually had small volumes of watery diarrhea, with a median number of diarrhea per day of six (range from four to twelve), and only one case caused by CNTP isolate showed typical "rice-water" stool [1]. Most patients vomited less than three times a day, and fever was reported in only 16% of cases after early infection. The overall dehydration was usually less than 10% of the body weight (S2 Table). In contrast, more than half (5/8) of the cholera toxin-producing L2 isolates caused severe dehydration and electrolyte abnormalities. Furthermore, infection with L3b and L9 strains occurred mostly in adults (20–50 years old) with no significant difference between genders (S7A Fig).

Both L3b and L9 strains caused disease over a long time period and were mainly isolated in summer when the mean temperature was over 20°C (S7B and S7C Fig). Different temporal patterns of incidence were observed. The L9 lineage only caused sporadic cases uniformly distributed over time, whereas an outbreak was observed for L3b cases in 2009. More specifically, the outbreak was mainly caused by the sub-lineage L3b.4, which first appeared in Hangzhou in 2006, after which the number of reported cases rapidly increased until its peak in 2009, and then quickly decreased in 2010 (S7D Fig), without any specific infectious disease control measure having been taken.

## Virulence factors, antimicrobial resistance and unique genomic variants in L3b and L9 isolates

We investigated the sequences of 15 previously well-described pathogenicity genomic islands [43] (CTXφ, VPI-1, VPI-2, VSP-1, VSP-2 and GI-1~10), the SXT and superintegron regions, and searched against VFDB (virulence factor database) [44] and ARDB (antibiotic resistance genes database) [45] to identify possible virulence and antibiotic genes in L3b and L9 isolates. Most of the L3b and L9 isolates (96%, 133/139) were CNTP, whereas few isolates (5%, 39/828) of the other lineages were CNTP (S1 Fig). Specifically, we found that 46% of L3b isolates and 15% of L9 isolates carried a precursor CTXφ [19] (pre-CTXφ), which is an incomplete CTXφ lacking the *ctxAB* genes (Figs 1B, 1C, S3A and S3B). Pre-CTXφ is considered to be the precursor of the current Cla and ET CTXφ, and it is found in many clinical and environmental isolates [19,52]. We found that the pathogenicity island VSP-1 was carried exclusively by L3b.4 isolates (Figs 1B and S3A). VSP-1 has been found to improve the efficiency of intestinal

colonization with *V. cholerae* [53] and was inferred to be an essential factor that led lineage L2 to cause the seventh pandemic [54].

We also found two genes outside of genomic islands that had an interesting distribution among isolates of L3b lineage. First, 70% of L3b isolates carried the quinolone resistance gene *qnrS*, with an even higher proportion of 83% among Hangzhou isolates (Fig 1B). *qnrS* was first reported in isolate MS6 from Thailand [55] (L3b.4), located in the superintegron region, which functions as a gene capture system with gene content varying considerably among *V. cholerae* isolates. Outside of the L3b and L9 lineages, the gene was only found in 4% isolates. The product of *qnrS* gene has been shown to protect gyrase and topoisomerase IV from quinolone action in enteric bacteria [56], however, most of the L3b and L9 lineage isolates are sensitive to commonly used antibiotics (S3 Table), including fluoroquinolones (ciprofloxacin and norfloxacin acid). Second, the virulence gene *acfD* was found exclusively in L3b.2 (Figs 1B and S3E), which is an accessory colonization factor and regulated by the same system that controls expression of the cholera toxin [57].

In addition, we found many unique variations of the investigated genome elements (Figs 1B, 1C and S3). Most L3b.1 isolates carried VPI-2 (S3C Fig) and GI-5 variants (S3F Fig), and half of the L3b.2 isolates carried a unique VSP-2 variant (S3D Fig), which is different from the ET-VSP-2 and WASA variants reported previously [5]. All the L9 isolates carried VPI-1, GI-5 and GI-10 variants (S3G–S3I Fig). Among the VPI-1 island, they carried *tcpA* variants from cluster 5 and 11, which are commonly found in environmental isolates [14].

## The local epidemiology of L3b and L9 lineages in Hangzhou

The geographical location of sampling or patient home addresses was recorded for 91 Hangzhou L3b and L9 isolates. The 81 L3b isolates showed a broad geographical distribution, which covered both urban area and rural districts, whereas ten L9 isolates originated almost exclusively from the urban area (Fig 2A). Compared to the distribution in the urban areas where all the sub-lineages mixed together, the rural isolates are clustered by geography, all the isolates from the southeast rural area belonged to L3b.2, and in contrast most northeast rural isolates were L3b.4 isolates. For both sub-lineages, the first infectious diarrhea cases were found in rural districts. The first patient infected with L3b.2 isolate was identified in 2001 in the southeast rural district, while the first urban L3b.2 patient was identified two years later (S8A Fig). The first rural patient infected with L3b.4 isolate was identified two months earlier (June and August) than urban patients. Furthermore, the phylogenic trees indicate that the rural isolates are generally located on deeper branches: the average root-to-tip node distance of the rural isolates was significantly lower than for urban isolates (5.7 vs. 8.3, *P* <0.01, Student's *t*-test, S8A Fig), which suggests that the urban isolates are most likely to have been imported from rural districts.

The epidemiological investigation collected four L3b and three L9 environmental strains. They were isolated from aquatic food products (soft-shelled turtle and bullfrog), water and tableware of a restaurant that had been in contact with the patients (Fig 2B and 2C and S1 Table). The phylogenetic trees revealed that these environmental isolates were closely related to the human isolates (Fig 2B and 2C): the median number of SNPs between any environmental isolate and its closest human isolate was 2 (range from 0 to 23). Two environmental isolates that had been isolated from a restaurant (water for keeping fish and tableware) showed only 0–2 SNP differences across the whole genome with isolates from four humans (including two patients and two carriers) who had eaten in the restaurant (Fig 2B); in contrast, the pairwise SNP distance between human isolates of L3b and L9 lineages were 55 and 127, respectively. This suggested that the contaminated aquatic food products caused infections through food

served in the restaurant. This is supported by the epidemiological investigation that these four people had eaten aquatic food products in this restaurant. In addition, epidemiological investigation found the source of six aquatic food products was in shopping malls and agricultural markets (S1 Table), suggesting that commercial markets can be a transmission route of L3b and L9 isolates.

## Evidence for the global spread of L3b and L9 lineages

In addition to the Hangzhou isolates, there were 46 L3b and L9 isolates from Turkmenistan, Russia, Ukraine, Thailand, Mexico, Haiti, Argentina and Paraguay (Fig 1B and 1C and S9, S1 Table). They were isolated from patients (48%) or environment (52%, water and food) from 1965 to 2015, during the diarrheal/cholera outbreaks or by routine surveillance [8,17,18,23–25,55] (S1 Table and S7B Fig). Due to the usually mild symptoms of human infection caused by these two lineages, their sampling is limited compared to the pandemic lineage, which can affect the inference of the routes and directions of spread. Given this sampling bias, we only aimed to construct the initial global picture of the epidemiology of these two lineages. We found a significant temporal signal in the L3b lineage (S5 Fig), which allowed us to estimate the approximate divergence times of the branches (S6 Fig). Based on the phylogeny, we observed that at least nine international spread events (ISEs) occurred (Figs 1B and S9) since the most recent common ancestor (MRCA) of the L3b lineage in 1926 (95% confidence interval: 1897–1950).

Our phylogeny shows that the introduction of L3b lineage into Hangzhou took place on at least three independent occasions (Figs 1B and S9). One introduction of L3b.1 sub-lineage (ISE1) occurred between 1995 and 2010 (the first date is the MRCA of local isolates and their closest relative from the source, the second date is the MRCA of local isolates only), which caused two human infections. The most closely related isolate to Hangzhou isolates was an environmental (water) isolate from Russia. These two Hangzhou isolates and the Russian isolate carried the unique GI-5 variant, which was only found in Hangzhou, Turkmenistan and Russia. The second import (ISE3) occurred between 1938 and 1970, which established the L3b.2 sub-lineage in Hangzhou. The oldest strain of this sub-lineage was isolated in Turkmenistan in 1965. The third import (ISE8) occurred between 1956 and 1997, which led to the establishment of lineage L3b.4 in Hangzhou, with three Ukraine strains being their closest relatives. These results indicate the close relationship between Hangzhou and global isolates, given that the closest global isolates (Turkmenistan, Russia and Ukraine) were located on more ancient branches (Figs 1B and S8B), with an average root-to-tip node distance of 4.8 vs. 9.8 (Hangzhou), the spread direction of L3b lineage was likely from these countries to Hangzhou. However, the genetic distance between Hangzhou and global isolates suggested that intermediates might be present between them, and therefore we cannot exclude the possibility that L3b lineage from other locations was the direct source of import into Hangzhou. Our phylogeny also revealed two possible export events from Hangzhou to other countries. The first was an export to Mexico (ISE4, 1988–1994) which seeded the L3b.2 sub-lineage circulating in Latin American countries. Although some Mexico strains were isolated earlier than Hangzhou strains (S7B Fig), all of them in L3b lineage were the descendants of Hangzhou isolates, indicating the sampling in Hangzhou or China is inadequate. The second was an export to Thailand (ISE9, 2005–2008), and one human clinical case was found to be a consequence of this export [55].

Two strains located nearest to the root of the phylogeny of L3b strains had both been isolated from a Central Asia country Turkmenistan, and these were the oldest isolates (both sampled in 1965) in this lineage, suggesting that the isolates of nearby countries could be imported from Central Asia. This is consistent with the epidemiological investigation which concluded

that Russia and Ukraine isolates had been imported from Central Asia, where the climate is warmer and more suitable for *V. cholerae* growth and persistence [25]. The global spread route of L3b strains are therefore probably originated from Central Asia, and then spread to the nearby countries (e.g. ISE2). As early as in the 1930s, the L3b strains spread to China (ISE 3, 1938–1970), and in about 1990 (ISE 4, 1988–1994) this lineage crossed the Pacific Ocean to arrive in Latin America where it has been present since.

L9 isolates were also found in Mexico and Haiti, and the ML tree suggests that these are also the descendants of the Hangzhou L9.1~2 sub-lineages (Figs 1A, 1C and S1). Because of limited sample numbers and short interval of sampling time in this lineage (all were collected after 1999), we could not detect a significant temporal signal and estimate dates (S5B Fig). Nevertheless, the phylogeny still indicates a close relationship between Hangzhou and Latin American isolates, and therefore L9 is another *V. cholerae* lineage that has spread globally.

## Discussion

Using whole genome sequencing, we identified L3b and L9, two serogroup O1 *V. cholerae* lineages that are distinct from the pandemic lineage (L2) and can cause infectious diarrhea. Our results indicate that the absence of *ctxAB* genes in these lineages is due to either carrying pre-CTXφ or totally missing the whole prophage sequence. Despite the absence of cholera toxin genes, the two lineages cause many infections over long time periods in Hangzhou, Russia and Latin America, indicating that they had established stable environmental reservoirs in many regions around the world. The human infections caused by L3b and L9 lineages correspond with the second pattern of disease proposed by Domman et al [8], i.e. a lineage causing a long-term disease across wide geographical regions, but with fewer human cases than the pandemic *V. cholerae* lineage, similar to the US Gulf Coast lineage which led to 65 seafood-borne disease cases from 1973 to 1992 [59].

Although the clinical symptoms caused by these two lineages in Hangzhou are usually milder than for the pandemic L2 lineage, they still showed considerable pathogenic potential. The CTXφ negative strains have been reported to be able to integrate CTXφ into their genomes and become cholera toxin producers [19]. This is supported by the fact that five L3b isolates carried a complete CTXφ (Figs 1B, S1 and S3), which shapes potential risk for cholera outbreak. Both lineages carried the TCP cluster, suggesting the ability to effectively colonize the human intestine, and recent experiment have verified that *V. cholerae* can colonize on the soft-shelled turtle [60]. This might associate with the global spread of L3b and L9 isolates, because they might be able to use aquatic animals or human as the medium. Pre-CTXφ could also be a virulence factor. Its virulence has been validated in an infant rabbit model [61], and the two phage proteins *zot* and *ace* are considered as the most probable virulence genes because of their enterotoxic activity [62]. Furthermore, other pathogenicity islands, such as VSP-1 in sub-lineage L3b.4, and unknown virulence factors may also be related with the virulence of these two lineages, which will need to be clarified in further work.

Foodborne disease outbreaks cause by non-toxigenic *V. cholerae* were reported in many developing and developed countries [15], such as the US Gulf Coast outbreak [59]. The human infections in Hangzhou are also likely to be foodborne and linked to aquatic food products. Seafood and aquatic food products contamination with *V. cholerae* have been confirmed in China by an epidemiological investigation, 80% of the isolates belonged to serogroup O1 and O139 [63]. The average contamination rate was 0.52% in 12,104 samples, and the most common contamination was found in turtles and their breeding pools (1.72% and 1.14%) [60]. In this study, a total of 71 clinical diarrhea cases were confirmed in Hangzhou over 12 years, which is on a similar scale as the US Gulf Coast outbreak [59]. Aquatic food products

consumption is very popular in Hangzhou, especially in the summer season, which is when the L3b and L9 infection cases emerged. We have isolated L3b and L9 strains from contaminated aquatic food products in a restaurant and commercial markets, and we showed that these environmental isolates have a close relationship with patient isolates based on a whole-genome phylogeny. This provides evidence that the aquatic food products were very likely to be the source of the human infections in Hangzhou. Furthermore, Hangzhou human infection cases are mostly distributed in adults, which may be because aquatic food products are mostly consumed by adults.

An interesting geographical pattern in the incidence of L3b and L9 infection in Hangzhou is that the epidemic seems to originate from an eastern rural area, where the aquatic food products breeding bases are located. The aquatic food products breeding pools could be the environmental reservoir of *V. cholerae*, from which contaminated food would be exported during harvest season, which would explain the long-lasting infections observed in Hangzhou. Therefore, our observations suggested that further epidemiological investigation was needed on these breeding bases. L3b and L9 infections may also occur in other regions in China with similar aquatic food products breeding bases, but as the *ctxAB* negative *V. cholerae* strains are usually considered to have a low risk of disease, such infections were possibly neglected during surveillance. The disease burden caused by these two lineages in other countries may also be underestimated.

China has previously been found to play an important role in the international spread of the seventh cholera pandemic, acting both as the source and the sink in many international transmission events [64]. Here our results suggest a similar role in the spread of other lineages of *V. cholerae*. The L3b strains was introduced into China on at least three occasions, and in turn, China was the source of L3b strains imported into Thailand and Latin America. International transmission was also observed in the L9 lineage. However, L3b and L9 isolates from Latin America were previously considered as local isolates (MX-2 and ELA-3, respectively) [8], when Chinese genomes were unavailable. Therefore, our results show that these lineages have larger population sizes and geographical ranges than previously thought. More thorough and unbiased sampling of *V. cholerae* is needed to reveal a more complete picture of its global epidemiology.

## Supporting information

**S1 Table. Background information of 969 *V. cholerae* isolates used in this research.**
(XLSX)

**S2 Table. The criteria used for clinical symptom classification.**
(PDF)

**S3 Table. Antimicrobial resistance of *ctxAB* negative, *tcpA* positive (CNTP) isolates.**
(PDF)

**S4 Table. Summary of the Bayesian models used for BEAST analyses.**
(PDF)

**S1 Fig. Maximum likelihood tree of 969 *Vibrio cholerae* isolates.** Branch colors indicate the lineages, and red circles in the tree tips indicate Hangzhou isolates. L3b and L9 lineages are highlighted after zooming in, and FastTree support values of key nodes are shown (range from 0 to 1). BAPS clustering results, *ctxAB* negative, *tcpA* positve (CNTP) isolates (red color) and the BLASTN coverage of *tcpA* and CTXφ genes (*ctxB*, *ctxA*, *zot*, *ace*, *orfU*, *cep*, *rstA*, *rstB*, *rstR*)

are shown on the right.
(TIF)

**S2 Fig. ClonalFrameML recombination analysis of L3b (A) and L9 lineages (B).** Left: ClonalFrameML reconstructed phylogeny. Right: dark blue horizontal bars indicate detected recombination events, grey areas indicate non-core regions. Two chromosomes are separated by dotted lines.
(TIF)

**S3 Fig. Unique genome elements of L3b and L9 isolates.** (A) Maximum likelihood trees of L3b (top) and L9 (bottom) as in Fig 1B and 1C are shown on the left. BLASTN coverage of 17 pathogenicity islands and genomic islands are shown on the right. The sequences of these genome elements were split into segments of 500 bp that corresponding to the columns on the plot. (B-I) ACT plot of genomic elements variants of L3b (top) and L9 isolates (bottom).
(TIF)

**S4 Fig. Maximum likelihood trees and hierarchical clustering results of L3b (A) and L9 (B) lineages.** Phylogenies and genome elements distributions are the same as Fig 1B and 1C, hierBAPS clustering results are shown on the right.
(TIF)

**S5 Fig. Assessment of the temporal signal of L3b and L9 lineage.** (A, B) Correlation between root-to-tip distance and isolation time of L3b and L9 (B) isolates. (C) Three date-randomization tests of L3b lineage.
(TIF)

**S6 Fig. BEAST maximum clade credibility tree of L3b isolates.** Colors of branches and tip labels indicate geographical locations. Circles in the tree tips indicate sources, red for clinical isolates, blue for carrier, green for environmental isolates and star for rural isolates. Inferred date and 95% confidence intervals of key nodes are labeled on the tree.
(TIF)

**S7 Fig. Clinical characters (A) and temporal distribution (B-D) of L3b and L9 isolates.** Colors indicate lineages, and circle size scales with strain number in (B, D). The line in (C) indicates the monthly mean temperature in Hangzhou, corresponding to the Y axis on the right.
(TIF)

**S8 Fig. Root-to-tip node number distribution of L3b isolates.** (A) Time distribution and root-to-tip node number distribution of rural (red) and urban (blue) isolates. (B) Root-to-tip node number distribution of isolates from different countries.
(TIF)

**S9 Fig. Inferred global spread routes of L3b and L9 isolates.** Circles and triangles indicate clinical and environmental isolates, respectively, and their sizes scale with sample number. Red and green colors indicate L3b and L9 isolates, respectively. Dashed lines indicate the inferred international spread event (ISE) routes. Blue circles indicate ever reported *ctxAB* negative clinical samples. The dates of spread events are taken from BEAST analysis, the first date is the median MRCA of transmitted isolates and their closest relative from the source, the second date is the MRCA of transmitted isolates only. The map was created using ggmap based on the public geographical data downloaded from OpenStreetMap.
(TIF)

## Author Contributions

**Conceptualization:** Jingcao Pan, Yujun Cui.

**Data curation:** Haoqiu Wang, Zhou Sun, Wei Zheng, Wei Zhang, Hua Yu.

**Formal analysis:** Chao Yang, Yujun Cui.

**Funding acquisition:** Ruifu Yang, Jingcao Pan, Yujun Cui.

**Investigation:** Haoqiu Wang, Zhou Sun, Wei Zheng, Wei Zhang, Hua Yu, Jingcao Pan.

**Methodology:** Chao Yang, Yarong Wu, Yujun Cui.

**Project administration:** Ruifu Yang, Jingcao Pan, Yujun Cui.

**Resources:** Haoqiu Wang, Zhou Sun, Wei Zheng, Wei Zhang, Hua Yu, Yujun Cui.

**Software:** Chao Yang, Yarong Wu.

**Supervision:** Ruifu Yang, Jingcao Pan, Yujun Cui.

**Visualization:** Chao Yang.

**Writing – original draft:** Chao Yang, Yujun Cui.

**Writing – review & editing:** Chao Yang, Xavier Didelot, Yujun Cui.

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
