## [Decision Letter · Decision Letter 0]

17 Aug 2019

Dear Dr. Cui:

Thank you very much for submitting your manuscript "Genomic epidemiology of Vibrio cholerae reveals the regional and global transmission of two epidemic non-toxigenic lineages" (#PNTD-D-19-00871) for review by PLOS Neglected Tropical Diseases. Your manuscript was fully evaluated at the editorial level and by independent peer reviewers. The reviewers appreciated the attention to an important problem, but raised some substantial concerns about the manuscript as it currently stands. These issues must be addressed before we would be willing to consider a revised version of your study. We cannot, of course, promise publication at that time.

We therefore ask you to modify the manuscript according to the review recommendations before we can consider your manuscript for acceptance. Your revisions should address the specific points made by each reviewer. 

When you are ready to resubmit, please be prepared to upload the following:

(1) A letter containing a detailed list of your responses to the review comments and a description of the changes you have made in the manuscript.

(2) Two versions of the manuscript: one with either highlights or tracked changes denoting where the text has been changed (uploaded as a "Revised Article with Changes Highlighted" file); the other a clean version (uploaded as the article file).

(3) If available, a striking still image (a new image if one is available or an existing one from within your manuscript). If your manuscript is accepted for publication, this image may be featured on our website. Images should ideally be high resolution, eye-catching, single panel images; where one is available, please use 'add file' at the time of resubmission and select 'striking image' as the file type. 

Please provide a short caption, including credits, uploaded as a separate "Other" file. If your image is from someone other than yourself, please ensure that the artist has read and agreed to the terms and conditions of the Creative Commons Attribution License at http://journals.plos.org/plosntds/s/content-license (NOTE: we cannot publish copyrighted images). 

(4) If applicable, we encourage you to add a list of accession numbers/ID numbers for genes and proteins mentioned in the text (these should be listed as a paragraph at the end of the manuscript). You can supply accession numbers for any database, so long as the database is publicly accessible and stable. Examples include LocusLink and SwissProt.

(5) To enhance the reproducibility of your results, we recommend that you deposit your laboratory protocols in protocols.io, where a protocol can be assigned its own identifier (DOI) such that it can be cited independently in the future. For instructions see http://journals.plos.org/plosntds/s/submission-guidelines#loc-methods

While revising your submission, please upload your figure files to the Preflight Analysis and Conversion Engine (PACE) digital diagnostic tool, https://pacev2.apexcovantage.com/ PACE helps ensure that figures meet PLOS requirements. To use PACE, you must first register as a user. Then, login and navigate to the UPLOAD tab, where you will find detailed instructions on how to use the tool. If you encounter any issues or have any questions when using PACE, please email us at figures@plos.org.

We hope to receive your revised manuscript by Oct 16 2019 11:59PM. If you anticipate any delay in its return, we ask that you let us know the expected resubmission date by replying to this email.

To submit a revision, go to https://www.editorialmanager.com/pntd/ and log in as an Author. You will see a menu item call Submission Needing Revision. You will find your submission record there. 

Sincerely,

Benjamin Althouse

Associate Editor

Mathieu Picardeau

Deputy Editor

Reviewer's Responses to Questions

**Key Review Criteria Required for Acceptance?**

**Methods**

-Are the objectives of the study clearly articulated with a clear testable hypothesis stated?

-Is the study design appropriate to address the stated objectives?

-Is the population clearly described and appropriate for the hypothesis being tested?

-Is the sample size sufficient to ensure adequate power to address the hypothesis being tested?

-Were correct statistical analysis used to support conclusions?

-Are there concerns about ethical or regulatory requirements being met?

Reviewer #1: (No Response)

Reviewer #2: Overall, the aim of this study is clear and the sampling strategy — testing of all suspected cholera cases admitted to hospitals in China — was clear and appropriate. I have two relatively minor comments related to the selection and presentation of methods in the manuscript:

1- It would be helpful if the authors would present a few more details about how the samples were prepared for sequencing in the lab (the paragraph starting at line 154). Whole genome sequencing of cholera is still relatively new, and publishing detailed methods that were used to successfully produce whole genomes for analysis would be a helpful contribution to the field. In particular, there was no mention of the DNA extraction kit used and the DNA concentration used as input in the extraction protocol. Additionally, there are many Illumina protocols for preparing paired-end sequencing libraries (Nextera Flex, Nextera XT, etc.) - which one was used in this paper?

2- It is unclear to me why the authors chose to present a neighbor-joining tree (rather than a maximum likelihood tree) in Figure 1a. Was there additional information gained from comparing the neighbor-joining tree produced in 1a to the maximum likelihood tree in Figure 1b? It should be computationally feasible to construct a maximum likelihood tree of the 969 isolates if using only SNP sites in the alignment, as stated in the Results.

Reviewer #3: I’m curious why you chose to assemble and then align using MUMmer, instead of mapping raw reads (which tend to be much more accurate in terms of calling variants) particularly for the within lineage analysis. See discussion here: https://doi.org/10.3389/fgene.2015.00235

Line 151: The BioProject is not public yet on NCBI.

Line 158: Can you provide the range of the number of contigs and sizes? What was the fold coverage of these genomes?

It would be helpful to provide the alignment lengths for all of the mapping analyses performed, for instance the total number of SNPs in the mapping back to N16961. 

Line 172: The 257,757 SNPs were called against which reference?

Line 172 -173: This is very confusing to follow. I would suggest re-arranging the sentence so that you report the SNPs per lineages together; ie 40,533 SNPs for the 119 L3b (27,006 SNPs for the 81 Hangzhou L3b isolates), etc…

When you are referring to ‘core genomes’ do you mean the MUMmer alignment against a reference? Core genome is more typically a term in a ‘pan-genome’ analysis to describe genes that are conserved in all or nearly all (~95%) genomes. 

What program was used to scale all of the ML trees into SNPs per site?

My biggest concern with the manuscript is how many sites were lost after applying ClonalFrameML to the datasets. For example, the alignment for L3b goes from 40,533 SNPs to 535 SNPs!! Yikes. As Xavier is an author on this paper, I would like to have his opinion on what is going on here. 

Are the original SNPs distributed across the genome or do they mostly appear in clusters? If in clusters, perhaps looking at some of the reads mapped back to these regions might give an indication of what is going on. 

Using the core-gene alignment from the Latin American paper, two strains (TUC_VC182 & 2284) within MX-2 have 749 SNPs between them. Looking at your Fig 1b, the scale bar would indicate there are less than 10-20 SNPs between any MX-2 isolates. It would be helpful to see the pre-CFML sub-trees of L3b and L9 to judge relatedness.

I would suggest that the authors use some sort of clustering algorithm (such as hierarchical BAPS) to guide how they designate lineages. It seems that there is tremendous diversity in just the L3b clade, to the point where I would be very hesitant to label them the same (sub)lineage.

For the root-to-tip distances, can you please load the L3b tree into TempEst and provide me the plot showing the ancestor traces please? This can help in identifying possible isolates that might be problematic. See here for further details: http://beast.community/tempest_tutorial. It does seem that if you did not include the four L3b isolates sampled prior to 1990, there might be very little signal.

Given that the R^2 values are not very good for the root to tip data, I’m going to be a pain and ask that the authors do a date randomization test to further confirm that the temporal signal in the data is real. 

Did you only run a single MCMC chain for each model in BEAST? Typically, one runs a minimum of three as to avoid getting stuck in local maxima. It might make sense to test other tree priors, such as non-parametric models like the Skyride or Skygrid as it may be that the L3b population has not remained stable over time. 

For the BEAST analysis, did the authors correct for constant sites by adding in these values to the XML files as described here: https://groups.google.com/forum/#!searchin/beast-users/SNPs/beast-users/V5vRghILMfw/jMtC_DwS5EYJ

**Results**

-Does the analysis presented match the analysis plan?

-Are the results clearly and completely presented?

-Are the figures (Tables, Images) of sufficient quality for clarity?

Reviewer #1: (No Response)

Reviewer #2: Figures 1-3 were clear, easy to understand, and supported the conclusions described in the text. While Figure 4 is clear and provides a nice visual explanation of the conclusions described in “Evidence for the global spread of L3b and L9 lineages”, I am generally uncomfortable with the visualization of arrows showing movement based on a small number of samples. The results are described nicely and with the appropriate caution (i.e. lines 344-345) but I find the solid line arrows on the map potentially misleading. For example, the evidence presented that Mexico strains descended from Hangzhou strains is convincing, but how do you know the transmission was direct from Hangzhou to Mexico, without a (possibly unrecorded or minimally symptomatic) intermediate? I would suggest moving this figure to the supplemental material, to avoid continuing the pattern in the cholera field of presenting tentative transmission events as hard facts.

Another small comment: it would be helpful if the authors could provide the number of SNPs in their full SNP alignments, both to add context to their comments about the similarity between human and environmental isolates (line 308) and to allow for comparison to previous studies of global cholera transmission. This could be provided in methods (around line 172) or when describing the results (line 221).

Reviewer #3: Of the 850 genomes included in the study, 540 of them were from 7PET (L2). This seems unnecessary as there was no analysis of these strains in the paper.

Line 220-230: How were lineages assigned and what criteria was used to designate or differentiate lineages? Same question for the sub-lineages. It seems L4 is a catch all for many different lineages, for instance there are 6 lineages from the Latin American paper that are within what you call L4. 

Line 244: Was the one case that showed classic rice water stool CPTP?

Line 300: Are these tip-to-root node distances based off of the recomb-free SNP alignment only or the pre-CFML alignment?

Line 321-323: Awkward sentence, consider re-wording. 

Global spread of lineages: I think the authors go too far in here in associating these lineages to global spread. As stated above, the loss of so many sites made these strains artificially appear very similar to those sampled in LA and elsewhere. Again, looking that the NJ tree in FigS1, there is more diversity within L3b than seen between L7 and L1. Also true for diversity seen between any lineages at the top of the tree (L5,L8,L6 & L2). There is one long branching isolate from China that is at the base of the MX-2 lineage (your L3b.2), which does not appear to have high bootstrap support. It does seem that ITE9 is plausible, likely due to shipment of contaminated food products, as this strain sits well within those from Hangzhou. 

General questions:

Do the authors think that there is secondary transmission (ie person to person)? Or rather is this a point source / intermittent common source of food borne illness? 

I bring this up as the words ‘transmission’ and ‘introduction’ are used in the manuscript. Perhaps I have misunderstood their meaning, but I take this to mean human to human transmission, which I am not convinced is the case here. 

Figure 1: Need to state that the b & c panels are ML trees from alignment post CFML. Also, the branch color for Paraguay is very difficult to see in the tree. You might consider changing this to a darker color. 

I would suggest moving Figure 2 to the supplement as well as panel D from Figure 3 -- maybe add this to Fig S5. 

Fig S1: Can you please label on the tree the other L3b sub-lineages as you have done with L3B.1.

**Conclusions**

-Are the conclusions supported by the data presented?

-Are the limitations of analysis clearly described?

-Do the authors discuss how these data can be helpful to advance our understanding of the topic under study?

-Is public health relevance addressed?

Reviewer #1: (No Response)

Reviewer #2: The conclusions are in general clear and well supported/caveated. However, there are two conclusions I feel are not entirely supported by the data that I would suggest the authors address:

1- In line 254, the authors claim the incidence rate of L3b cases revealed temporal fluctuation. However, I would argue that it’s possible that they happened to see an outbreak of this lineage during the sampling time frame, and that it’s possible the other lineage caused an outbreak outside of the 2001-2012 time period. It may be more appropriate to state that an outbreak was observed for L3b cases, and not try to draw larger conclusions from this observation.

2- In line 294, the authors comment that it is interesting to see that isolates from the southeast rural area belonged to L3b.2 and ones from the northeast were L3b.4. I don’t find this particularly surprising or remarkable, since it would be expected that lineages would cluster somewhat by geography. I have no problem with the authors describing their observations, I just don’t find it particularly noteworthy.

Reviewer #3: This is nice work describing outbreaks caused by non-pandemic VC lineages. I agree with the authors that we need a much better sampling of non-pandemic VC. 

The link to sea turtles as potential transport mechanisms seems a bit too tenuous at this stage.

**Editorial and Data Presentation Modifications?**

Reviewer #1: (No Response)

Reviewer #2: The manuscript was generally well-written and easy to follow. A few small comments:

1- The paragraph starting in line 77 is an abrupt transition from the previous paragraph’s discussion of lineages. A better transition to the topic would be helpful.

2- The sentence staring in line 85 is confusing to read. Rewording this sentence would be helpful and could also be used to help transition into this section (see previous comment).

3- The large arrows in Figure 3a are a bit distracting, and are not necessary to convey the authors’ conclusions.

4- It is more common to see “root-to-tip” than “tip-to-root” distance (e.g. line 343 and Fig S5). Perhaps this could be changed for consistency.

Reviewer #3: Line 237: Please define FSU countries at first use. I would suggest dropping the use of FSU however, and either just state the countries specifically referred to or use the UN or WHO regional grouping names.

**Summary and General Comments**

Reviewer #1: Comments on manuscript PNTD-D-19-00871 entitled " Genomic epidemiology of Vibrio cholerae reveals the regional and global transmission of two epidemic non-toxigenic lineages"

In this manuscript, the authors describe genomic features of Vibrio cholerae O1 isolates 

associated with 71 atypical cholera cases in Hangzhou, China between 2001 and 2012. The data generated by this study are interesting as they add knowledge on these atypical non-toxinogenic Vibrio cholerae O1 isolates in terms of clinical symptoms (mild disease) and epidemiology (probable foodborne infection caused by contaminated seafood or aquatic products). Furthermore, this study shows that the MX-2 lineage previously thought to be geographically restricted to Latin America (Domman et al. Science 2017) is globally distributed. 

It would have been, however, interesting to know more on the antimicrobial resistance phenotype and resistome of these Chinese CNTP isolates. Maybe there is no other AMR gene than qnrS. In this case it is worth to be mentioned. For the Chinese isolates containing the qnrS gene within the superintegron (Lineage 3b.4), what is the sensitivity to nalidixic acid and ciprofloxacin?

Reviewer #2: “Genomic epidemiology of Vibrio cholerae reveals the regional and global transmission of two epidemic non-toxigenic lineages,” describes the analysis of 119 CNTP isolates from Hangzhou. The authors find that these isolates belong to two lineages, and use previously-published cholera sequences to contextualize their transmission on a global scale. They provide evidence for multiple transmission events of these lineages to and from China, and advocate for more widespread surveillance of these non-toxigenic lineages. These findings are well-described with important caveats about sampling, and clearly show that these lineages are being transmitted outside of Latin America — the only place they have been previously reported. With some minor edits, this manuscript could make a nice contribution to the available literature on non-toxigenic lineages, and would add to the growing collection of cholera whole genome sequences. By providing a few more details on how the samples were prepared (see my comment in the Methods section above), the manuscript can also become a resource for others attempting to sequence cholera isolates.

Reviewer #3: Here the authors have sequenced 119 VC genomes isolated in Hangzhou, China over a period of time stretching between 2001 to 2012. Their results show that many of these non-pandemic strains ‘fill out’ lineages within the larger Vibrio cholerae species tree and will provide a really great resource for future studies. Once we have addressed some of the major concerns I raised above, I would be happy to see this manuscript published.

PLOS authors have the option to publish the peer review history of their article (what does this mean?). If published, this will include your full peer review and any attached files.

Reviewer #1: No

Reviewer #2: No

Reviewer #3: Yes: Daryl Domman

---

## [Decision Letter · Decision Letter 1]

10 Dec 2019

Dear Dr. Cui:

Thank you very much for submitting your manuscript "Genomic epidemiology of Vibrio cholerae reveals the regional and global spread of two epidemic non-toxigenic lineages" (PNTD-D-19-00871R1) for review by PLOS Neglected Tropical Diseases. Your manuscript was fully evaluated at the editorial level and by independent peer reviewers. The reviewers appreciated the attention to an important topic but identified some aspects of the manuscript that should be improved.

We therefore ask you to modify the manuscript according to the review recommendations before we can consider your manuscript for acceptance. Your revisions should address the specific points made by each reviewer.

(1) A letter containing a detailed list of your responses to the review comments and a description of the changes you have made in the manuscript.

(2) Two versions of the manuscript: one with either highlights or tracked changes denoting where the text has been changed (uploaded as a "Revised Article with Changes Highlighted" file ); the other a clean version (uploaded as the article file).

(3) If available, a striking still image (a new image if one is available or an existing one from within your manuscript). If your manuscript is accepted for publication, this image may be featured on our website. Images should ideally be high resolution, eye-catching, single panel images; where one is available, please use 'add file' at the time of resubmission and select 'striking image' as the file type. 

Please provide a short caption, including credits, uploaded as a separate "Other" file. If your image is from someone other than yourself, please ensure that the artist has read and agreed to the terms and conditions of the Creative Commons Attribution License at http://journals.plos.org/plosntds/s/content-license (NOTE: we cannot publish copyrighted images). 

(4) Appropriate Figure Files 

Please remove all name and figure # text from your figure files upon submitting your revision. Please also take this time to check that your figures are of high resolution, which will improve both the editorial review process and help expedite your manuscript's publication should it be accepted. Please note that figures must have been originally created at 300dpi or higher. Do not manually increase the resolution of your files. For instructions on how to properly obtain high quality images, please review our Figure Guidelines, with examples at: http://journals.plos.org/plosntds/s/figures

While revising your submission, please upload your figure files to the Preflight Analysis and Conversion Engine (PACE) digital diagnostic tool, https://pacev2.apexcovantage.com/ PACE helps ensure that figures meet PLOS requirements. To use PACE, you must first register as a user. Then, login and navigate to the UPLOAD tab, where you will find detailed instructions on how to use the tool. If you encounter any issues or have any questions when using PACE, please email us at figures@plos.org.

We hope to receive your revised manuscript by Feb 08 2020 11:59PM. If you anticipate any delay in its return, we ask that you let us know the expected resubmission date by replying to this email.

To submit your revised files, please log in to https://www.editorialmanager.com/pntd/

Sincerely,

Benjamin Althouse

Deputy Editor

Mathieu Picardeau

Deputy Editor

Reviewer's Responses to Questions

**Key Review Criteria Required for Acceptance?**

**Methods**

-Are the objectives of the study clearly articulated with a clear testable hypothesis stated?

-Is the study design appropriate to address the stated objectives?

-Is the population clearly described and appropriate for the hypothesis being tested?

-Is the sample size sufficient to ensure adequate power to address the hypothesis being tested?

-Were correct statistical analysis used to support conclusions?

-Are there concerns about ethical or regulatory requirements being met?

Reviewer #2: The methods section is much improved in response to the reviewer comments. A few other minor comments to consider:

1- The paragraph starting at line 108 and the paragraph starting at line 136 are largely redundant. Combining these into one clear and concise paragraph would be helpful.

2- It would be helpful to add a topic sentence to before line 121 to clarify the purpose of the next two paragraphs (determining which strains are CNTP). They are quite confusing otherwise.

3- Line 123: It is unclear that you are talking about published studies (timing of ctx loss) and not your own data. Please clarify.

4- Line 156: Why 850 genome sequences? When did you download the sequences and were there only 850 available at the time? Adding the date would be helpful, as there are now over 1200 available.

5- Line 205: I agree with Reviewer #3 that the drop from 18000 SNPs to a few hundred after using ClonalFrameML is striking. I appreciate you recombination rate calculations, but are there other software you could use to filter out recombination, so you could compare the results of the two? It's just such a big difference - would be good to check using another method.

Reviewer #3: (No Response)

**Results**

-Does the analysis presented match the analysis plan?

-Are the results clearly and completely presented?

-Are the figures (Tables, Images) of sufficient quality for clarity?

Reviewer #2: The results are much more straightforward in the revised manuscript, and I appreciate the toned-down approach to many of the tenuous conclusions.

Line 362: Would it be possible to provide the average number of SNPs between human cases in the various lineages? This would help contextualize the 0-2 SNP number.

Reviewer #3: Line 46-47: I think the "led to the establishment..." is a bit strong, as we still have only limited sampling of these lineages. Perhaps change to "are related to the groups..."

Line 52: frame  framework

**Conclusions**

-Are the conclusions supported by the data presented?

-Are the limitations of analysis clearly described?

-Do the authors discuss how these data can be helpful to advance our understanding of the topic under study?

-Is public health relevance addressed?

Reviewer #2: (No Response)

Reviewer #3: (No Response)

**Editorial and Data Presentation Modifications?**

Reviewer #2: (No Response)

Reviewer #3: (No Response)

**Summary and General Comments**

Reviewer #2: (No Response)

Reviewer #3: I thank the authors for the effort put forth in the revision. All of my comments were addressed and I look forward to seeing this publication in print.

PLOS authors have the option to publish the peer review history of their article (what does this mean?). If published, this will include your full peer review and any attached files.

Reviewer #2: No

Reviewer #3: No

---

## [Editor Report · Decision Letter 2]

9 Jan 2020

Dear Dr. Cui,

We are pleased to inform you that your manuscript, "Genomic epidemiology of Vibrio cholerae reveals the regional and global spread of two epidemic non-toxigenic lineages", has been editorially accepted for publication at PLOS Neglected Tropical Diseases.

Before your manuscript can be formally accepted and sent to production you will need to complete our formatting changes, which you will receive in a follow up email. Please note: your manuscript will not be scheduled for publication until you have made the required changes.

IMPORTANT NOTES

* Copyediting and Author Proofs: To ensure prompt publication, your manuscript will NOT be subject to detailed copyediting and you will NOT receive a typeset proof for review. The corresponding author will have one final opportunity to correct any errors when sent the requests mentioned above. Please review this version of your manuscript for any errors.

* If you or your institution will be preparing press materials for this manuscript, please inform our press team in advance at plosntds@plos.org. If you need to know your paper's publication date for media purposes, you must coordinate with our press team, and your manuscript will remain under a strict press embargo until the publication date and time. PLOS NTDs may choose to issue a press release for your article. If there is anything that the journal should know, please get in touch.

*Now that your manuscript has been provisionally accepted, please log into EM and update your profile. Go to http://www.editorialmanager.com/pntd, log in, and click on the "Update My Information" link at the top of the page. Please update your user information to ensure an efficient production and billing process.

*Note to LaTeX users only - Our staff will ask you to upload a TEX file in addition to the PDF before the paper can be sent to typesetting, so please carefully review our Latex Guidelines [http://www.plosntds.org/static/latexGuidelines.action] in the meantime.

Best regards,

Benjamin Althouse

Deputy Editor

Mathieu Picardeau

Deputy Editor

---

## [Editor Report · Acceptance letter]

10 Feb 2020

Dear Dr. Cui,

We are delighted to inform you that your manuscript, "Genomic epidemiology of *Vibrio cholerae* reveals the regional and global spread of two epidemic non-toxigenic lineages," has been formally accepted for publication in PLOS Neglected Tropical Diseases.

Best regards,

Serap Aksoy

Editor-in-Chief

Shaden Kamhawi

Editor-in-Chief
